# Centriole Translational Planar Polarity in Monociliated Epithelia

**DOI:** 10.3390/cells13171403

**Published:** 2024-08-23

**Authors:** Antoine Donati, Sylvie Schneider-Maunoury, Christine Vesque

**Affiliations:** 1Developmental Biology Unit, UMR7622, Institut de Biologie Paris Seine (IBPS), Sorbonne Université, CNRS, INSERM U1156, 75005 Paris, France; 2Department of Cell and Developmental Biology, University of California San Diego, La Jolla, CA 92093, USA

**Keywords:** planar cell polarity, translational polarity, epithelial polarity, cilia, basal body, centriole, Par3, Vangl2

## Abstract

Ciliated epithelia are widespread in animals and play crucial roles in many developmental and physiological processes. Epithelia composed of multi-ciliated cells allow for directional fluid flow in the trachea, oviduct and brain cavities. Monociliated epithelia play crucial roles in vertebrate embryos, from the establishment of left–right asymmetry to the control of axis curvature via cerebrospinal flow motility in zebrafish. Cilia also have a central role in the motility and feeding of free-swimming larvae in a variety of marine organisms. These diverse functions rely on the coordinated orientation (rotational polarity) and asymmetric localization (translational polarity) of cilia and of their centriole-derived basal bodies across the epithelium, both being forms of planar cell polarity (PCP). Here, we review our current knowledge on the mechanisms of the translational polarity of basal bodies in vertebrate monociliated epithelia from the molecule to the whole organism. We highlight the importance of live imaging for understanding the dynamics of centriole polarization. We review the roles of core PCP pathways and of apicobasal polarity proteins, such as Par3, whose central function in this process has been recently uncovered. Finally, we emphasize the importance of the coordination between polarity proteins, the cytoskeleton and the basal body itself in this highly dynamic process.

## 1. Introduction

Most epithelia display planar cell polarity (PCP), characterized by the coordinated positions and orientations of external structures on their apical cell surface, whether they are microtubule-based cilia or actin-based trichomes and stereocilia. PCP allows motile cilia to generate a directional flow of extracellular fluid, which is crucial, for instance, for proper cerebrospinal fluid (CSF) circulation or for the establishment of left–right asymmetry in vertebrate embryos [1]. PCP is also instrumental in organizing and orienting mechano-sensing structures such as kinocilia in zebrafish neuromasts or stereocilia bundles in the organ of Corti of the mammalian inner ear. Cilia within monociliated epithelia are built from the oldest centriole of the pair, the mother centriole [2] (Figure 1), while the vast majority of motile cilia from multi-ciliated cells are templated by daughter centrioles [3]. Besides this difference, the asymmetric positioning of cilia implies an asymmetric positioning of centrioles. To grow a cilium, a centriole will have to be modified and acquire distal appendages (DAs), which are protein complexes mediating centrioles anchoring either to the plasma membrane or to the ciliary vesicle (Figure 1) (reviewed in [4]). The modified centriole, called the basal body, is also decorated by sub-distal appendages. In motile cilia, one such appendage of triangular shape is called a basal foot and points in the direction of the ciliary stroke in multi-ciliated cells [5,6], as well as in monociliated cells [7]. Other sub-distal appendages called rootlets also decorate the basal body, but their exact number and orientation depend on the cell type and the maturation of the epithelium (reviewed in [8]) (Figure 1). The basal body nucleates ciliary microtubules that constitute the axoneme, protruding into the extracellular space.

After their initial docking near the center of the apical membrane [9,10,11], basal bodies move off-center in a coordinated manner within the plane of the epithelium (a process called translational polarity) and progressively align the orientation of their centrioles and appendages to control their beating angle and 3D movements (a process called rotational polarity) (Figure 2). 

These two forms of polarity appear coupled, as both of them depend on the function of the PCP pathway [12,13,14]. PCP has been extensively studied in *Drosophila* melanogaster, where it is controlled by a core of so-called “PCP proteins”. Interestingly, PCP proteins are widely conserved among metazoan [15,16] and play a critical role in basal body off-centering [17]. Here, we review the dynamics of basal body off-centering within the apical cell surface, a process called “translational polarization”, in several models of monociliated epithelia (Figure 3). We will focus on the key molecular actors of translational polarization, put forward an unforeseen role for other polarity proteins such as Par3 and question axonemal contribution in this process. We will describe the different cytoskeletal remodeling events allowing for centriole displacement in polarized epithelia from different animal models thanks to recent genetic and live imaging studies. These studies highlight an important contribution of the remodeling of the microtubule network, extending from the immediate vicinity of the basal body, either from the pericentriolar material involved in microtubule nucleation and/or from the basal body sub-distal appendages, and anchoring to specific membrane domains. Finally, we will highlight novel research directions that may bring valuable information to the field. 

However, it has proven technically challenging to perform live imaging in these systems, and the dynamics of both processes is still under-studied. A pioneering study using live imaging on cultured mouse embryos suggested that centrioles undergo a posterior shift between the tail bud and three somite stages (5 h) in the node [11] (Figure 4).

However, the low time resolution of this movie (one image every 30 min) made it difficult to draw definitive conclusions about centriole behavior in this system. Interestingly, the authors also pointed out that centriole translational polarization was asynchronous between node cells, which is similar to the results recently obtained for the zebrafish floor plate [18]. In the mouse cochlea, another study [19] showed using fixed samples that centriole planar polarization occurs between embryonic day (E) 15.5 and P0 (Figure 4). Live imaging performed on cochlear explants from E15.5, by taking one image every 15 min for up to 4 h, could not detect any translation of the centriole, which led the authors to propose a progressive and slow centrifugal movement for the centriole (17 nm/h in average). Thus, the limited data available so far in the mouse node and cochlea suggest that centrioles undergo a slow and progressive shift to the cell periphery. (Figure 4). 

This is in stark contrast with the conclusion of our recent study on translational polarization in the zebrafish floor plate. In this system, we uncovered fast back-and-forth planar apical movements of the centrioles at early developmental stages (4 to 8 somite stages), with an average speed of 0.2 µm/min, which is around 700 times faster than what has been proposed for the mouse cochlea (Figure 4). As somitogenesis proceeds, floor plate centrioles tend to spend more and more time in contact with the posterior apical membrane, where they eventually dock [18]. It will thus be very interesting to investigate centriole planar polarization dynamics in other species and organs to assess how common high centriole dynamics is compared to the relatively low dynamics proposed in mouse models. One promising system is the *Drosophila* pupal wing, where centriole translational polarization occurs within a 2 h timeframe between 30 and 32 h after puparium formation [20] (Figure 3 and Figure 4).

## 2. Role of Polarity Proteins in Centriole Translational Polarization 

### 2.1. Asymmetric Enrichment of Core PCP Proteins Controls Centriole Planar Polarization

PCP has been extensively studied in *Drosophila melanogaster* epithelia where its establishment relies on a set of core PCP proteins which localize asymmetrically at apical cell–cell junctions in a mutually exclusive fashion, with Vang Gogh (Vang) and Prickle (Pk) on one side and Frizzled (Fz), Dishevelled (Dvl) and Diego (Dgo) on the opposite side, while Flamingo (Fmi) interacts with both sides. These PCP proteins and their asymmetric localizations and roles in PCP establishment are conserved across metazoa [16].

The first evidence of a role of PCP proteins in the planar polarization of vertebrate ciliated epithelia came from the study of the mouse cochlea. In *Vangl2*^−/−^ mice (*Vangl2* is a *Vang* ortholog), PCP is disrupted in the cochlea. Depending on the position of the cell within the cochlea, these defects are either in the planar polarization process itself or in the coordination of this process between cells [10]. Vangl2 and Fz3 (a Fz ortholog) are both asymmetrically localized in cochlear cells, and Fzd3 localization depends on Vangl2 [21]. In this system, the cilium is off-centered at the opposite side of Vangl2 enrichment [21] (Figure 3 and Figure 4). In contrast, Dvl2 is asymmetrically enriched on the side towards which centrioles will move, as shown by immuno-labeling [22].

Similarly, in the mouse node, Vangl1 and Pk2 localize anteriorly [23], opposite to cilia in the off-centering direction, while a fusion Dvl2-GFP protein localizes at the posterior cortex, towards which basal bodies will become off-centered [11]. All of these PCP proteins are required for the planar polarization of the basal body. [*Vangl1*^−/−^; *Vangl2*^−/−^] mice display random cilia positioning in the node, leading to left–right asymmetry defects [24] (Figure 4). Pk1 and Pk2 are also required for translational polarization [25]. In addition, knocking out five of the six Dvl alleles (in [*Dvl1*^−/−^; *Dvl2*^−/−^; *Dvl3^+/^*^−^] compound mutants) leads to translational polarization defects, resulting in impaired directional flow in the node [11]. Thus, in both the cochlea and in the node, centrioles position towards the side of the cell enriched in Dvl, opposite to Vangl localization (Figure 4). 

Of note, the role of PCP in cilia positioning in the laterality organ is conserved in other vertebrates: in the Xenopus embryo gastrocoel roof plate, *Vangl2* MO-mediated knockdown disrupts cilia off-centering [23], while in the zebrafish Kupffer’s vesicle, maternal zygotic *Vangl2*^−/−^ embryos display abnormal flow [13].

In the zebrafish floor plate, Vangl2 is also enriched anteriorly, opposite to posterior cilia [13,26] (Figure 4). A recent study using endogenously tagged Vangl2-GFP confirmed its anterior localization. Maternal zygotic (MZ) *Vangl2*^−/−^ embryos display translational polarity defects in the floor plate [13]. Vangl2 anterior localization depends on the presence of Vangl2 in adjacent cells [27], a well-known feature of PCP proteins. The authors analyzed the consequences of the acute degradation of endogenous Vangl2 using the “zGrad” system and showed that Vangl2 is required in the posterior neighboring cell to keep the basal body at the posterior membrane at late larval stages [27]. Since Fz and Vang extracellular domains can mediate interactions between neighboring cells in *Drosophila* [28], the authors hypothesized that anterior Vangl2 in one floor plate cell could interact with Fz3a at the base of posterior cilia from its immediate anterior neighbor [27,29]. Indeed, they observed apical anterior membrane protrusions from a floor plate cell reaching the basal body of its immediate anterior neighbor, forming an oriented digitation. The properties and functions of these orientated membrane protrusions in translational polarization have not yet been assessed.

An important question that is not addressed by these studies is the following: what is the consequence of PCP defects on centriole behavior? Our live imaging study addressed this question in the zebrafish floor plate and revealed that polarization defects in *Vangl2*^−/−^ floor plate cells are coupled to basal body behavioral defects; *Vangl2*^−/−^ basal bodies still move with a similar speed to that in controls, but they make more frequent contacts with the lateral membranes than wild-type basal bodies [18]. This suggests that the role of PCP is to provide planar polarity cues at the apical membrane, which will then be used for the translational polarization of the basal body. 

The role of PCP in translational polarity is not limited to monociliated epithelia. Many studies in multi-ciliated cells have confirmed the asymmetric localization of PCP proteins and their role in centriole planar positioning: in Xenopus larvae epidermis [30,31,32], in mouse ependymal multi-ciliated cells and in their monociliated radial progenitors [12,33,34,35]. In non-vertebrate metazoa, fewer studies are available. Interestingly, in the *Drosophila* wing, in which PCP has been studied for many years, it was shown only recently that centrioles assume an asymmetric distal position, where they localize just beneath trichomes. This translational polarization is disrupted in Flamingo loss of function and Frizzled over-expression and completely reversed when over-expressing the Spiny-legs Prickle isoform [20]. The trichome is not a cilium, but in addition to actin filaments, it contains acetylated microtubules in proximity with the centrioles [20]. The presence of microtubules suggests that trichomes could be derived from highly modified cilia that would have lost the nine-microtubule doublet organization. Finally, in hydrozoan cnidarian planulae (free swimming larvae), PCP proteins are also involved in basal bodies/motile cilia orientation and positioning; in *Clytia hemisphaerica*, a *Vang* ortholog is required for the proper translational polarization of monociliated ectodermal cells to allow aborally directed swimming of the planula [36]. 

Together, these studies argue for a widely conserved role of PCP proteins in the orientation and positioning of cilia in mono- as well as multi-ciliated epithelia, with the cilia being off-centered towards the side of Dvl enrichment, opposite to Vangl2 enrichment at apical junctions (Figure 3 and Figure 4).

### 2.2. Apico-Basal Polarity Proteins Modulate PCP Proteins’ Distribution 

Since PCP proteins are localized at apical cell–cell junctions in epithelia, it seems obvious that apico-basal polarization, in which the PAR, Crumbs and Scribble complexes are involved, is a prerequisite for PCP protein asymmetric localization and PCP establishment. However, several studies have shown that it is not only a permissive requirement; the loss of function of some apico-basal polarity proteins can lead to PCP protein localization defects without having any noticeable effect on apico-basal polarity. 

In the *Drosophila* eye, which displays a striking planar polarization of ommatidial cells, the Crumbs complex member Patj binds Fz and limits its action, probably via aPKC, in a subset of ommatidial cells. The Par3 ortholog Bazooka (Baz) antagonizes the action of Patj and aPKC [37]. In the *Drosophila* wing, Baz over-expression does not affect apico-basal polarization but leads to a failure to restrict Fmi to the proximal and distal membranes, and Baz interacts directly with one of the two Fmi isoforms present in the wing [38]. Along the same line, Scribble1 is required for Par3 and Vangl2 apical localization in mouse neural tube cells, although Scribble1 mutants do not display severe apico-basal polarization defects [39].

In these systems, apico-basal polarity proteins play an upstream modulatory function on PCP protein localization. In addition, and more intriguingly, several components of the apico-basal polarity complexes (the Par, Crumbs and Scribble complexes) have been shown to behave as downstream effectors of PCP proteins. 

### 2.3. Role of Apico-Basal Polarity Proteins as Downstream Effectors of PCP

In *Drosophila*, polarity proteins are downstream effectors of PCP during asymmetric divisions of Sensory Organ Precursor cells (SOPs). In these cells, PCP proteins are required for the asymmetric localization of Par proteins along the antero-posterior axis [40]. It has recently been shown that Meru, a RASSF9/RASSF10 ortholog, is recruited to the posterior cortex by Fz/Dsh and recruits Baz/Par3 [41]. Vang colocalizes at the anterior cortex with the Scribble complex member Dlg, and Vang and Dlg recruit Pins. Pins and Dsh can both recruit Mud, which, in turn, recruits dynein to orient cell division along the antero-posterior axis [42,43,44] (Figure 4). In non-dividing cells, such as *Drosophila* eye cone cells, Baz/Par3 also displays a planar asymmetric localization downstream of PCP proteins [45].

In vertebrates, accumulating data involve the apico-basal polarity protein Par3 in centriole polarization. In the Xenopus embryo neural plate, Par3 is planar polarized downstream of Vangl2 [46], although in this case, it is not known whether it is anteriorly or posteriorly enriched. Reciprocally, Par3 interacts with Pk3 and is required for Vangl2 planar polarization. More recently, Par3 has been shown to be involved in cochlea PCP and basal body positioning [47]. Par3 has both a cell autonomous and non-cell autonomous effect, as it is required both for basal body positioning within the apical surface and for the coordinated orientation of cochlear cells. 

The most comprehensive results on Par3 function come from the study of the zebrafish floor plate [18]. Our live imaging study of the process uncovers a role for Par3 in translational polarization. As already described, in this system, basal bodies move back and forth between the anterior and posterior apical junctions where they make membrane contacts. We have found that contacts are made exclusively at Par3 local accumulations (“patches”). Par3 is enriched posteriorly prior to basal body movement, and disrupting Par3 localization (by its own over-expression or inactivation) leads to translational polarity defects. Furthermore, the posterior enrichment of Par3 patches is lost in *vangl2*^−/−^ floor plate cells. The fragmentation and spreading to lateral membranes of Par3 patches does not impair their ability to attract basal bodies but randomizes the positions of basal body contacts with apical junctions: basal bodies now make contacts with anterior, lateral and posterior membranes with the same probability. Thus, Vangl2 controls the asymmetric distribution of Par3 in zebrafish floor plate cells, which is crucial for correct basal body posterior positioning [18]. These results led us to propose that Par3 is a key link between tissue-level PCP (which depends on core PCP proteins like Vangl2) and intrinsic PCP (i.e., the translational polarization of basal bodies). We propose the following scenario: as Vangl2 localizes anteriorly in the floor plate [26,27], it could interact with Fz3a in its anterior neighbor [27,29]. Posterior Fz3a would then recruit Dvl, which would indirectly recruit Par3, enabling basal body attraction towards the posterior side. Available data suggest that the Dvl/Par3 interaction is probably indirect: in *Drosophila* SOPs, Dvl recruits Par3 to the posterior membrane via the RASSF protein Meru [41], while in the mouse cochlea, Dvl recruits Par3 via Daple, which colocalizes with Par3 and can bind both Dvl and Par3 in yeast two-hybrid assays [48].

## 3. “Ciliary” Proteins Modulate Translational Planar Polarity

Several proteins localizing at the base of cilia and/or along the axoneme (the part of the cilium protruding out of the apical surface), which are important for ciliogenesis or cilia function, have been shown to control basal bodies/cilia positioning in the mouse cochlea and ependymal multi-ciliated cells. However, it is now becoming clear that most “ciliary” proteins have extraciliary functions [49], which could be important for their role in the planar polarization of basal bodies. Thus, whether the cilium itself is involved in its own planar polarization is unclear. 

### 3.1. Role of BBS Proteins in Translational PCP

Bardet Biedl Syndrome proteins (BBS) are components of the BBSome complex that plays a critical role in removing proteins from cilia [50]. In the mouse cochlea, the depletion of the ciliary protein BBS6 (Mkks) or BBS4 leads to PCP defects; stereocilia bundles are misoriented or flattened [51], although the cilium itself (also called “kinocilium”) forms normally. Compound heterozygous mice for *BBS6* and *Vangl2* show defects in stereocilia similar to *BBS6* homozygous mutants, uncovering a genetic interaction between *BBS6* and *Vangl2*. In addition, Vangl2 was found to localize around basal bodies and within the axoneme in murine inner medullary collecting duct (IMCD3) epithelial cells and human multi-ciliated respiratory epithelial cells [51]. BBS4 localizes at centriolar satellites in HeLa cells and serves as an adaptor for dynein, allowing for PCM1 (a component of centriolar satellites) recruitment to the centriole and microtubule anchoring [52]. Thus, BBS4 and BBS6 may be important for Vangl2 localization to basal bodies, which could, in turn, be required for proper kinocilium positioning (note that the localization of Vangl2 to basal bodies has only been found in this study). BBS4 and BBS6 may also be required for the proper asymmetric cortical localization of Vangl2, or it may regulate the dynamics of microtubules emanating from the basal bodies to position them within the apical surface. Supporting the first hypothesis, another study showed that the ciliary proteins BBS8 and Ift20 (an IFT protein localizing dynamically at the trans-Golgi, at the cilium and along microtubules, required for vesicle trafficking from the trans-Golgi to the base of the cilium [53]) are required for proper Vangl2 asymmetric localization in the cochlea [54]. In support of the fact that some ciliary proteins might regulate Vangl2 protein distribution within the cells, a Bio-ID study identified Vangl2 protein in close proximity to two transmembrane transition zone proteins of the MKS complex, Tmem216 and Tmem17, located at the base of cilia [55].

### 3.2. Role of IFT Proteins in Translational PCP

Ciliary proteins can have an effect on PCP independently of PCP proteins, as suggested by two studies which showed that *Ift88* [56] and *Kif3a* [57] mouse mutants display cochlear PCP defects without the disruption of the asymmetric localization of PCP proteins. Ift88 and Kif3a (a subunit of Kinesin II) are two components of anterograde intraflagellar transport (IFT), which transports cargoes along axonemal microtubules from the base to the tip of cilia. They are essential for cilia formation and function. 

In *Ift88* mutant mice, the cochlea is shorter and wider, suggesting a role for *Ift88* in cochlear convergence and extension, a process in which PCP proteins are involved [10,22]. Most stereocilia bundles have a normal V-shape, but their orientation is not coordinated, which is reminiscent of *Vangl2* mutant polarization defects [10]. In addition, 10–15% of hair cells have a central kinocilium, surrounded by stereocilia, indicating defects in kinocilium migration to the cell cortex that are not seen in core PCP mutants. Finally, Vangl2’s and Fzd3’s asymmetric subcellular localization are not affected. This shows that Ift88 acts either downstream or in parallel to core PCP components to position the kinocilium at the cortex and to position it at the right place. Interestingly, the apical microtubule network emanating from the basal body is disrupted in Ift88^−/−^ cochlear cells, and Ift88 has been shown to regulate astral microtubule formation in mitotic cells [58] and to control spindle orientation in a PCP- and cilia-independent manner in zebrafish [59]. Thus, Ift88’s effect on basal body/cilium positioning in the cochlea could be linked to its ability to regulate the apical microtubule network. In zebrafish, maternal zygotic *ift88* mutants never form cilia and have no translational polarization defects in the floor plate [59]. Although this excludes a role for the axoneme in translational polarization, an extraciliary role of Ift88 remains possible since this mutant allele can still produce a truncated protein by exon-skipping [60], and basal bodies are still properly anchored to the apical membrane. 

*Kif3a* mouse mutant cochlea has no kinocilium and displays PCP defects such as reduced convergence extension [57]. In the middle and basal regions of the cochlea, stereocilia bundles are flattened, and there is a general loss of correlation between the basal body position and the middle of the stereocilia bundle. The off-centering of the basal body is not affected, but it is localized deeper within the cytoplasm compared to control hair cells. Intriguingly, the loss of Kif3a does not affect Dvl2 or Fzd3 asymmetric localization but disrupts the asymmetric localization of phosphorylated PAK, a kinase activated by Rac. Inhibiting Rac or PAK recapitulates the PCP defects seen in Kif3a mutants, although the kinocilium is not affected, suggesting that Kif3a acts through Rac/PAK independently of the axoneme to position the basal body in the cochlea. Kif3a could also play a role in cochlear PCP via an interaction with basal body-associated microtubules, since it localizes at basal body sub-distal appendages, which control basal body-associated microtubule nucleation [61].

### 3.3. Role of Rpgrip1l in Translational PCP 

Finally, *Rpgrip1l*, the master regulator of the ciliary transition zone (a region between the basal body and the axoneme which controls trafficking in and out of the cilium), is also involved in basal body/cilia positioning both in mice and zebrafish [62]. *Rpgrip1l* mutant mice have convergence extension and PCP defects in the cochlea, severely affecting kinocilium positioning while leaving stereocilia orientation nearly unaffected. The disconnection between the kinocilium and stereocilia bundle positions in the cochlea is a common feature of ciliopathy mutants [56] and is different from that of PCP mutants. In zebrafish, morpholino-mediated *rpgrip1l* knockdown leads to convergence extension defects as well as translational polarization defects in the floor plate, and this can be rescued by Dvl over-expression [62]. Accordingly, in Madin-Darby canine kidney (MDCK) cells, Rpgrip1l was shown to antagonize Inversin and Nphp4, two transition zone proteins that target Dvl for proteasome-mediated degradation [62]. Thus, Rpgrip1l could modulate the PCP pathway by regulating the stability of the core PCP protein Dvl. 

These examples suggest that several ciliary proteins play a role in basal body positioning that is independent of their role in axoneme formation and point to their role either in the intracytoplasmic transport of PCP components, in PCP protein stability, in the general organization of the microtubule network or in basal body connection with the subapical cytoskeletal network. Although the axonemal function of some ciliary proteins (Ift88, Kif3a, Pkd1 and Pkd2) is required to refine ciliary beating polarity or coordinate it at tissue level in ependymal multi-ciliated cells, most probably via the mechanosensory properties of cilia [34,63], the absence of a translational polarity defect in the maternal zygotic *ift88* zebrafish mutant floor plate rules out such a function in the translational polarity of this tissue [59].

## 4. Upstream Polarity Cues for PCP Orientation

In order to achieve physiologically adequate directional fluid flow generation, basal body planar polarization must be coordinated with the embryonic axis. We describe the known cues linking these global axes to PCP proteins and basal body positioning.

### 4.1. Wnt Ligands

Wnts are secreted glycoproteins that are specific to the metazoan lineage [64]. They have roles in cell differentiation and polarization, and it has been shown in cultured cells that a subset of Wnt ligands can both induce asymmetric cell fate and orient the mitotic spindle, which led to the hypothesis that Wnt allows the emergence of a coupling mechanism between these two key processes in metazoan development [64,65]. A potential link with the core PCP pathway came from the fact that Fz transmembrane proteins are Wnt receptors and that many Wnt ligands are expressed in a graded fashion; they are thus good candidates for providing a global cue for PCP orientation.

In *Drosophila*, it was long assumed that Wnt did not have any effect on PCP because individual Wnt mutants did not present PCP phenotypes in the wing. This assumption was challenged by the finding that Wg (one of *Drosophila* Wnts) and Wnt4 form a decreasing gradient from the wing margin, and that Wg or Wnt4 mis-expression causes PCP defects that are reminiscent of Fz’s loss of function. This suggests that Wg negatively regulates Fz, perhaps by competing with Vang for Fz binding. Indeed, Wg can prevent Fz binding to Vang in cultured *Drosophila* cells [66]. However, more recent studies [67,68] showed that the simultaneous loss of all endogenous Wnts does not lead to impaired PCP in the *Drosophila* wing, suggesting that a Wnt gradient is not necessary to orient PCP and that other cues (such as the Fat/Dchs system and mechanical forces; see below) are sufficient for PCP establishment. These other cues must somehow interact with Fz since Fz mutation leads to strong PCP defects [69].

In the mouse inner ear, Wnt5a is expressed in a gradient that is complementary to the gradient of the Wnt inhibitor Sfrp3 (Soluble Frizzled Related Protein 3) and is required for proper cochlea elongation via convergence extension [70]. In addition, Wnt5a cooperates with Vangl2 (a Vang ortholog) to properly orient sensory hair cells [70].

A recent study further showed that the conditional KO of Wntless (which prevents Wnt secretion) in the cochlea leads to convergence extension defects, the uncoupling of the basal body and stereocilia and tissue-level PCP disruption [71] without affecting basal body movement towards cell junctions, a phenotype reminiscent of core PCP mutants. Interestingly, Fz6 and Dvl2 localization were disrupted, but not Vangl2 or Dvl3, suggesting that there might be additional cues upstream of some PCP core proteins’ localization.

In the mouse embryonic node, in which planar polarization is required for left–right axis establishment [11], Wnt5a and Wnt5b are expressed on the posterior side of the node (Figure 4), whereas the Wnt inhibitors Sfrp1,2 and 5 are expressed on its anterior side. Wnt ligands and Sfrps are required for core PCP components’ asymmetric localization and basal body off-centering. Interestingly, a uniform expression of the Wnt ligands or their antagonists cannot rescue the absence of these molecules, demonstrating an instructive role of Wnts and their inhibitors for the node’s antero-posterior planar polarization [72].

In contrast, in the zebrafish floor plate, Jussila et al. [27] found no evidence of a role of exogenous Wnt5b or Wnt11 on basal body posterior positioning, but this negative result is to be taken with caution as neither the amounts of exogenous Wnt5b and Wnt11 nor the effect on the endogenous gradient could be monitored due to the lack of specific antibodies against these ligands. Nevertheless, these studies suggest that Wnt proteins can have an instructive role in global PCP both in vertebrates and *Drosophila*.

### 4.2. Fat/Dchs Protocadherins

Fat and Dachsous are protocadherins that were first found to regulate the growth of *Drosophila* imaginal discs via the Hippo pathway [73]. It was later discovered that Ft and Ds have a role in PCP in *Drosophila*. Like core PCP module components, Ft and Ds are asymmetrically localized in the cells of planar polarized epithelia and are required for PCP coordination at the tissue scale. In *Drosophila* wings, Ds is expressed in a decreasing proximo-distal gradient. Conversely, there is a decreasing Disto-proximal gradient of Four-jointed (Fj) expression; Four-jointed is a Golgi-resident kinase that can phosphorylate both Ft and Ds, but with opposing effects. Ft phosphorylation by Fj leads to an increased affinity of Ft for Ds in adjacent cells, whereas Ds phosphorylation by Fj leads to a decreased affinity of Ds for Ft. Thus, the Fj gradient produces a decreasing disto-proximal Ft affinity gradient that is complementary to the Ds gradient [74]. The interaction of these gradients results in the asymmetric localization of Ft and Ds on opposite sides of epithelial cells [75]. The Ft/Ds gradients can orient sub-apical non-centrosomal microtubules along the polarity axis, and it has been shown that the core PCP component Fz can be transported distally in *Drosophila* wing cells along such microtubules [76]. In *Drosophila*, pupal wing Ft and Ds are required for centriole off-centering, similar to what was previously reported for Fz-PCP alterations [77]. Thus, it appears that in this system, these two pathways are non-redundant.

In vertebrates, the role of the Ft/Ds/Fj system in PCP is much less clear, although some studies have shown a role of Fat4 (a Ft ortholog) in PCP-related processes [78,79]. Nothing was known about a potential role of this system in basal body planar positioning until recently, when Sai et al. [25] demonstrated that Dchs1 and Dchs2 (the orthologs of *Drosophila* melanogaster Ds) have a role in basal body posterior positioning in mouse embryonic node pit cells. Dchs1/Dchs2 double mutant embryonic node pit cells have defective Vangl1 and Vangl2 asymmetric localization, and the basal body fails to become posterior. These results suggest a conserved role of the Ft/Ds/Fj system in basal body planar positioning in bilaterians. 

### 4.3. Mechanical Forces

Many ciliated epithelia experience fluid flow at their surface, and it has been found that this flow can establish or refine the orientation and position of the basal body and cilia.

It was first shown that in Xenopus embryonic epidermis multi-ciliated cells, an initial bias in ciliary beating polarity, which initially depends on a planar polarization of basal body appendages, is in a second phase refined by fluid flow. Fluid flow can even reorient cilia polarization, but only when cilia are motile, suggesting a positive feedback mechanism, where flow aligns ciliary beating, which then reinforces the flow [80]. A subsequent study on mouse ependymal multi-ciliated cells [34] showed that fluid flow can establish coordinated ciliary beating polarity in wild-type mice but not in *Vangl2* mutants or in *Kif3a* mutants lacking cilia, suggesting a model in which fluid flow initiates a positive feedback loop that also requires Vangl2. In this system, the mechano-sensitive channels Pkd1 and Pkd2 might transduce this mechanical signal since they are only expressed along the axonemes of multi-ciliated cells and their mutation triggers a loss of ciliary beating polarization [63].

More recently, a study in which forces were exerted with a micropipette on Xenopus gastrocoele roof plate explants to mimic mechanical constraints during gastrulation showed that a directional mechanical tension can trigger the asymmetric enrichment of core PCP proteins and the asymmetric positioning of basal bodies/cilia and even modulate the length of cilia in these monociliated cells, although the molecular mechanisms at work in that context are still unknown [81]. Fluid flow could also play a role in mouse node polarization, but it is unlikely to be required for zebrafish floor plate polarization since the opening of the neural tube central canal takes place after polarization is complete [18,29]. However, this does not preclude mechanical constrains exerted by neighboring cells during convergence extension and the posterior elongation of the axis to provide such information.

## 5. Control of Basal Body Movements Downstream of Polarity and Ciliary Proteins

The molecular mechanisms triggering basal body movements within the apical surface are still poorly understood, but progress has been made in recent years, shedding light on the role of cytoskeletal elements such as actin filaments and microtubules.

### 5.1. Role of Actin and Myosin

In mouse ependymal multi-ciliated cells, non-muscle myosin II (NMII) B and C localize close to basal bodies. In this system, NMII is required for translational polarity [33]. NMIIB and NMIIC also localize at the cell cortex, and it is therefore difficult to conclude whether NMII has a role in translational polarization specifically at basal bodies. However, two pieces of data, the exclusive localization of phosphorylated Myosin Light Chain (pMLC), a marker of active NMII, at basal bodies and the impairment of translational polarity by inhibiting MLC phosphorylation by MLC Kinase (MLCK) strongly reinforce that NMIIB and C have a role at the basal body in triggering its movement. 

Active NMII is also required for basal body off-centering in mouse embryonic node pit cells [25] (Figure 5).

In these cells, pMLC localizes at the anterior cell cortex downstream of PCP proteins, and MLCK inhibition (with a different inhibitor, ML7) impairs basal body off-centering (Figure 5). Furthermore, Rac1 inhibition (with NSC23766) impairs pMLC anterior localization and basal body off-centering [11,25], suggesting that Rac1 activates NMII to achieve proper basal body off-centering. Interestingly, these pharmacological inhibitions do not impair Vangl1 anterior localization, which shows that Rac1 and MLCK activate NMII downstream of PCP proteins to position the basal body posteriorly [25].

Disrupting the actin filaments themselves also impairs basal body/centriole apical off-centering. In mouse embryonic node pit cells, stabilizing the actin network with Jasplakinolid inhibits basal body off-centering, which could be explained by a steric inhibition of basal body movements by a rigid actin apical network. Conversely, inhibiting actin polymerization with Cytochalasin D in *Drosophila* pupal wing impairs PCP, as centrioles are dispersed randomly all over apical surfaces [82]. Thus, in this cellular context, the actin network is required to keep the centriole in a confined region of the apical surface: in the middle before polarization and at the distal end after polarization (Figure 5). These results suggest that actin has a permissive role for centriole off-centering in this system and that another mechanism, downstream of PCP proteins, is involved in triggering centriole directional movement, potentially involving microtubules. However, a role of differential local actin dynamics within the apical surface cannot be ruled out and it will be important to investigate actin dynamics and centriole movements through live imaging of these cells.

The link between basal bodies and actin could be mediated by so-called ciliary adhesions, composed of adhesion proteins FAK (Focal Adhesion Kinase), Paxillin and Vinculin, which have been shown to localize next to basal bodies in Xenopus larvae epidermis multi-ciliated cells and connect basal bodies to actin to allow for their migration, docking and spacing within the apical surface [83]. 

### 5.2. Role of Microtubules

Centrioles/basal bodies are microtubule-based structures at the core of centrosomes, which are classically described as one of the main microtubule-organizing centers of animal cells [84]. It is therefore likely that basal body-associated microtubules help generate the forces that lead to basal body planar off-centering. However, proving this direct mechanical role experimentally promises to be challenging since microtubules are also involved in the asymmetric distribution of PCP proteins, which are themselves involved in basal body planar off-centering.

#### 5.2.1. The Role of Microtubules in the Localization of PCP Proteins

Subapical, non-centrosomal microtubule arrays oriented along the planar polarity axis have been found in many planar polarized epithelia [85,86,87]. In *Drosophila* pupal wings, these microtubules allow for the directional trafficking of Fz-, Fmi- and Dsh-containing vesicles [76,88,89], which could serve to amplify asymmetry or provide the initial polarity bias by removing proximal Fmi-Fz-Dsh complexes and transporting them to the distal side. Microtubules are also required for the asymmetric localization maintenance of Vangl2 in zebrafish floor plate cells [29] and of Vangl1 in mouse embryonic node pit cells [25]. In contrast, microtubules seem dispensable for the asymmetric localization maintenance of Pk in zebrafish embryonic mesoderm [90], of Vangl2 in mouse oviduct multi-ciliated cells [91] and of Pk2 and Vangl1 in mouse tracheal multi-ciliated cells [85]. Reciprocally, PCP proteins have been shown to be required for microtubule polarization, suggesting the existence of a feedback loop between microtubule orientation and PCP protein asymmetric localization [85,88,92]. Finally, some polarity proteins’ apical localization depends on microtubules, such as Par3 apical localization in zebrafish neural tube cells [93].

#### 5.2.2. Direct Role of Microtubules in Basal Body Movement

Despite the reciprocity in function and the resulting difficulties in data interpretation, several studies strongly suggest that microtubules could also have more direct roles in basal body planar orientation and positioning.

In the mouse cochlea, EB1 (a microtubule (+) end binding protein) is enriched at the cortex, close to the kinocilium final position [94] (Figure 5). The same study also found a weak enrichment of dynein at this cortical site, suggesting a mechanism of microtubule pulling forces at the cortex, leading to basal body/kinocilium off-centering. Interestingly, in this system, basal body/kinocilium positioning depends on the mInsc/LGN/Gαi polarity proteins that are involved in oriented cell division. It was hypothesized that these proteins could act on basal body positioning by regulating astral microtubule dynamics [94], but there is currently no clear demonstration of potential cortical microtubule capture by mInsc/LGN/Gαi and Dynein that would pull the basal body to the cell periphery [95].

In the zebrafish floor plate, our recent study suggests a microtubule-based mechanism for translational polarity. Live imaging revealed the existence of EB3 comets (marking the (+) ends of microtubules) traveling from the basal body towards its target membrane shortly (10 s) before basal body movement [18] (Figure 5). In this system, basal body movement toward the target membrane is often associated with a local bending or digitation of the membrane toward the basal body (Figure 5), which suggests the existence of a microtubule-mediated pulling force between the basal body and the membrane. These membrane digitations have been observed in other cases of centriole off-centering, such as in the oriented division of epidermal cells in ascidian embryos [96] or in immune cell polarization [97], for which microtubule (+)-end capture-shrinkage at the cortex has been proposed to generate a pulling force on the centrioles.

In the mouse embryonic node, a recent study found asymmetry in basal body-associated microtubules before basal body off-centering; anterior-pointing basal body-associated microtubules are organized horizontally within the apical surface, whereas posterior basal body-associated microtubules display a more pronounced vertical tilt (Figure 5). This could generate an asymmetry in the forces exerted by microtubules on the basal body, leading to its posterior positioning, although the authors acknowledge that the nature of these forces remain mysterious [25].

#### 5.2.3. Role of Daple and Par3 in Microtubule-Mediated Basal Body Polarization

Recent studies in mouse ependymal multi-ciliated cells point towards a direct role of microtubules and dynein in basal body planar positioning via Daple [98]. Cortical dynein, anchored at the anterior cell cortex via Daple, is required for translational polarity, probably via a microtubule pulling mechanism. In mouse tracheal multi-ciliated cells, Daple KO impairs the planar polarized apical microtubules and disrupts basal body planar orientation without affecting core PCP proteins. This phenotype is consistent with a direct interaction of Daple with microtubules and with its microtubule bundling and stabilizing activities [99]. In the mouse cochlea, Daple colocalizes with Par3 and interacts with Dvl and Par3 in yeast two-hybrid assay [48], suggesting that it might mediate dynein-mediated basal body translational polarization.

Par3 can also interact directly with dynein for centrosome orientation in migrating cells [100] or indirectly in *Drosophila* neuroblasts where Par3 recruits Dynein via Inscutable and Mud to orient cell division [101] (Figure 4). Moreover, Par3 can trigger microtubule catastrophe at the cortex by inhibiting Trio, a Rac-GEF, during the contact inhibition of locomotion in migrating neural crest cells [102]. These results suggest that Par3 could anchor basal body-associated microtubules at the cortex and locally regulate their dynamics to generate pulling forces on the basal body to trigger and maintain its planar off-centering.

### 5.3. Links between Basal Body Appendages and the Cytoskeletal Network during Polarization

In multi-ciliated cells, basal bodies present on their distal portion several appendages that connect with the cytoskeletal network, including, notably, a basal foot pointing in the direction of ciliary beating (Figure 1 and Figure 2). Gamma-tubulin localizes to basal feet in human oviduct ciliated cells [103] and tracheal cells, where it promotes and stabilizes a microtubule network that runs both subapically and as pillars along the apicobasal axis [104]. In this cell type, the apical microtubule network links basal feet to membrane domains associated with Fz and Dvl to orient cilia [85]. Moreover, in Xenopus multi-ciliated cells, basal feet-associated microtubules link basal bodies to their neighbors to promote the coordination of cilia orientation [105]. Finally, a striking observation in the tracheal multi-ciliated cells of *Odf2* hypomorphic mutant mice was reported: basal bodies that are properly docked on the apical membrane have specifically lost their basal feet and display cilia orientation defects. The asymmetric expression of PCP proteins is preserved, but the apical cytoskeleton organization is abnormal [6,106]. The role of basal feet and Odf2 in cilia orientation and their connection to microtubules are conserved in planarian flatworms [107]. Overall, basal feet are crucial for rotational polarity and for the polarized organization of the apical cytoskeleton network.

Very little is known in motile monociliated cells about a potential role of basal body appendages in organizing a polarized microtubule network for translational polarity. In non-motile cilia, a prominent striated rootlet may be present as in photoreceptors [108], while smaller sub-distal appendages may vary in number depending on cell type [109] or be present as one to six small conical structures in endothelial cells [110]. In motile monociliated cells, one might expect to observe a single sub-distal appendage that would behave as a basal foot oriented in the direction of cilia beating. A basal foot is clearly evidenced by a dense triangular structure on one side of the basal body that points in the direction of ciliary stroke in the case of swimming sea urchin embryos that are covered by monociliated ectodermal cells [7]. Another study showed the presence of planar polarized gamma-tubulin rootlets at the base of monocilia in swimming gastrulating embryos of the medusa *Clytia Hemispherica,* which very likely correspond to oriented basal feet [36]. It would therefore be very interesting to assay the subcellular localization of Odf2 in monociliated planar polarized cells, as it is required for basal foot formation in multi-ciliated cells [6,106,107] to test whether subdistal appendages are asymmetrically distributed around the monocilia basal body and whether this potential rotational polarity precedes the process of translational polarity.

Interestingly, recent data show that the basal body-associated proteins vfl1/LRRCC1 and C2CD3 are asymmetrically distributed around the basal bodies of non-motile monocilia in human retinal pigmented epithelial cells, exhibiting the same type of asymmetrical distribution as the one observed in polarized ependymal cells [111].

In ependymal cells, LRRCC1 and C2CD3 appear rotationally polarized around the basal body opposite of the basal feet. These asymmetrically positioned proteins can maintain basal body orientation along the PCP axis and behave as anchoring landmarks for basal body appendages [107]. It would therefore be of great interest to perform conditional loss of function of Odf2 and Vfl1 in polarized monociliated cells; one might expect to prevent the correct formation of basal body appendages without affecting the distribution of PCP proteins and to be able to assay both translational and rotational polarities. Would both types of polarities be coupled in this specific genetic context? Would the microtubule network be modified? Experimentally answering these questions may allow us to address translational polarity dependence on polarized basal body appendages and associated cytoskeletal structures.

## 6. Conclusions and Perspectives

Uncovering how polarity proteins and cytoskeletal fibers interact with the basal body during the highly dynamic process of translational polarization is a fascinating challenge. Genetic experiments addressing polarity or cytoskeletal protein function during translational polarity establishment are challenging and require their activity and localization to be manipulated with exquisite spatio-temporal resolution. Since the pioneer model of PCP, *Drosophila melanogaster*, does not present planar polarized cilia, there is room for vertebrate models such as mouse and zebrafish models. Indeed, this last model is amenable to genetics, live imaging and optogenetic manipulation; some recent studies suggest that it would be possible to dynamically manipulate the subcellular localization of polarity proteins, such as Par3 [112], or to perform acute loss of function using the Z-GRAD system [27] while assessing the effects on basal body movements. These approaches will be essential to improve the understanding of the dynamic processes leading to cilia planar polarization and of the respective roles of polarity proteins, basal body appendages and the cytoskeleton.

Recent data from several laboratories and on several model systems identified Par3 as an important actor of cilia translational polarization. Par3, as well as its interactor Daple, emerges as an important intracellular effector that controls microtubule distribution via its interactors to orient the polarity axis downstream of PCP function. Of note, Par3 can attract centrioles in different cellular contexts, even in the absence of PCP, such as centriole migration to the presumptive apical side of the cell and the concomitant reorganization of the microtubule network [113,114]. During zebrafish neurulation, Par3 accumulates on the membranes of forming apical cellular junctions [114]. In *Drosophila* male germ cells, Par3 local enrichment allows for the attraction of centrioles on membrane junctions expressing high levels of E-cadherin [115]. We can therefore speculate that in tissues where the Wnt-PCP pathway is active, patches are formed by Par3 recruitment by Dishevelled and its interactors (such as Daple or Meru/RASSF7) to attract centrioles on apical junctions enriched in Dvl. The recruitment of Par3 as a downstream effector of translational polarity has been evidenced in *Drosophila* SOPs [40], in mouse cochlea [47] and in zebrafish floor plate cells [18]. This suggests that Par3 could play a more global role in centriole attraction.

As the Wnt-PCP pathway is broadly used during embryo morphogenesis and regulates a number of cellular processes besides cilia polarization, mutations of the Wnt-PCP effectors Par3 and Daple, involved in cilia polarization, are expected to trigger a subset of Wnt-PCP phenotypes, which could overlap with the ones found in the case of defective ciliary motility.

Rare heterozygous variants in Vangl1 or Vangl2 are associated with various neural tube defects (NTDs) in humans, including spina bifida. Interestingly, several of these variants were found to behave as dominant or semi-dominant alleles for PCP phenotypes when expressed in *Drosophila* [116]. So far, only one study reported an association between rare heterozygous variants in PAR3/PARD3 and NTDs [117], and it remains to be tested whether these variants indeed alter the PCP pathway. Furthermore, one could predict that defective cilia polarization will trigger abnormal inner fluid dynamics that could phenocopy mutations in ciliary motility genes, leading to primary ciliary dyskinesia (PCD) in humans. It is worth noting that several recessive truncating mutations in the Daple/CCDC88C gene, encoding DVL and PAR3 interactors, have been found in fetuses with congenital hydrocephalus and is therefore called Congenital Hydrocephalus-1 or HYC1 [118]. This defect partially mimics the phenotype of some heterozygous mutations found in the PCD gene FOXJ1 or CILD43 (with FOXJ1 being the master gene of ciliary motility), which associates congenital hydrocephalus with respiratory problems, and in 50% of cases, laterality problems [119]. Since CCDC88C is expressed very strongly in brain ependymal cells, this might explain the restriction of the phenotype to the central nervous system. It is worth noting that CSF flow in humans does not heavily rely on ependymal cilia motility [120], contrary to the situation observed in mice and zebrafish with smaller brain cavities.

Indeed, cilia and CSF flow are crucial in juvenile zebrafish to maintain the polymerization of the Reissner Fiber, a CSF polymer of the SCO-SPONDIN protein, that runs from the anterior brain to spinal cord posterior tip and that maintains a straight axis in that species. The conditional degradation of SfGFP-Vangl2 in zebrafish Foxj1 lineages leads to juvenile scoliosis, a phenotype similar to that of hypomorphic mutations in the sco-spondin gene (sspo) itself [27,121,122]. In agreement with their phenotype, conditional *vangl2* juveniles present defective Reissner Fiber formation and ectopic SCO-SPONDIN aggregates within central nervous system cavities. A truncating mutation in one of the four *par3* genes in zebrafish, *pard3ab*, also leads to a juvenile scoliotic phenotype that has not been investigated yet (*pard3ab* ^fh305^ mutant line [123]). Overall, these data strengthen the idea that PAR3/PARD3 and DAPLE/CCDC88C may also behave as Wnt-PCP effectors in humans, presenting a subset of Wnt-PCP phenotypes, especially within the central nervous system.

The involvement of cilia-oriented beating in directional fluid flow and organism locomotion is a highly conserved process throughout evolution. Are the molecular mechanisms ensuring this coordination conserved? Since Wnt-PCP is required for cilia planar positioning in the cnidarian *Clytia hemisphaerica* [36], it would be interesting to test whether Par3 could be involved as a PCP effector in this species, with cnidarians being a sister group of bilaterians. Par3 sub-cellular localization in *Clytia hemisphaerica* has not been reported yet, while a *Che-Par3* gene is present in their genome (Momose, unpublished). In another metazoan branch, the ctenophores, the Wnt-PCP pathway is not conserved [15]. In the comb jellies belonging to this group, the local coordination of ciliary beating and the orientation of the giant comb cilia appear to depend on newly discovered proteins that are only present in Ctenophores and participate in the formation of physical connectors between adjacent cilia named compartmenting lamellae [124,125]. Cilia beating coordination may thus have been achieved through different strategies among metazoans, and it will be important to investigate the involvement of Par3 in the control of centriole planar polarization in *Clytia hemisphaerica* to determine whether its role in cilia polarization downstream of the Wnt-PCP pathway is a bilaterian-specific feature or if it also encompasses cnidarians.

## Figures and Tables

**Figure 1 cells-13-01403-f001:**
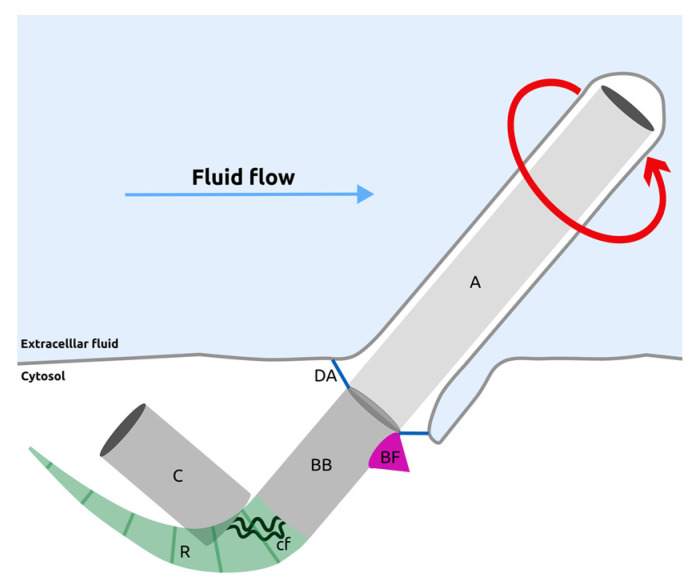
A motile cilium from monociliated epithelia and its appendages. The axoneme (A) of the cilium is composed of nine radially organized doublets of microtubules, extending from the basal body (BB) and a central doublet of microtubules in most motile cilia (note that in the monocilia of the mouse node, both 9 + 2 and 9 + 0 formulas are found). The basal body is connected to the plasma membrane by distal appendages (DAs), is linked to a duplicated centriole (C) by connecting fibers (cf) an displays polarized sub-distal appendages. **The basal foot (BF) is on the side of ciliary beating, while the rootlets (R) are most often on the opposite side of the basal foot.** These appendages connect the cilium to the cell cytoskeleton, ensuring its mechanical stability and proper positioning. Ciliary beating is indicated by a red circular arrow, and fluid flow direction by a blue arrow.

**Figure 2 cells-13-01403-f002:**
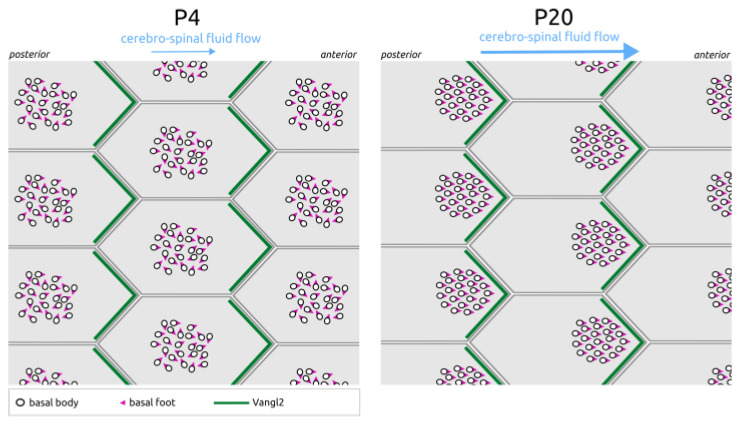
Cilia translational and rotational PCP in mouse ependymal multi-ciliated cells. Between post-natal day 4 (P4) and P20, the mouse ependymal multi-ciliated cells undergo two forms of planar cell polarization (PCP). Rotational PCP refers to the uniform orientation of cilia basal feet in the direction of the cerebro-spinal fluid flow, while **translational PCP refers to the off-centering of all cilia** on the anterior side of the apical cell surface. Both types of polarization depend on PCP proteins, such as Vangl2, whose asymmetric anterior localization is shown here in green.

**Figure 3 cells-13-01403-f003:**
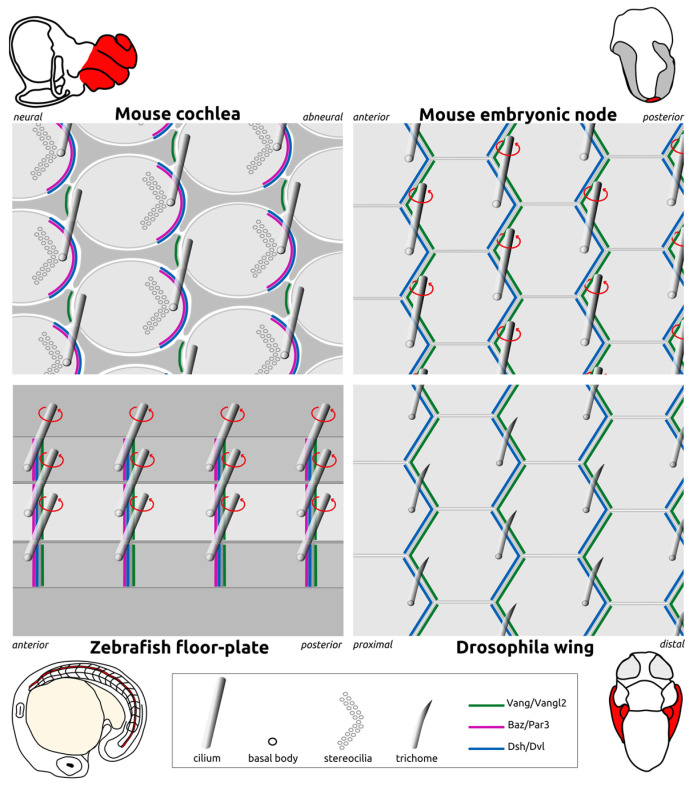
Examples of planar polarized monociliated epithelia. Planar polarized monociliated epithelia display **a coordinated off-centering of cilia on one side of the apical cell surface, enriched in Dsh/Dvl**. This coordinated polarization relies on the asymmetric localization of core PCP proteins (Vang/Vangl2; Dsh/Dvl) and other polarity proteins such as Baz/Par3. The top (apical) views of epithelia are shown: the organ of Corti in the adult mouse cochlea (top left; the cochlea is colored red on the top schematic of the mouse newborn inner ear), the mouse embryonic node (top right; shown in red on the top schematic of a E7.5 embryo), the zebrafish floor plate (bottom left; medial floor plate in light gray and lateral floor plate in dark gray, with the floor plate colored red on the schematic of the 18 somite-stage embryo below) and the *Drosophila* pupal wing (bottom right; the pupal wings are colored red on the schematic of a pupa 33 h after puparium formation). The *Drosophila* pupal wing epithelium is not ciliated but displays the translational polarization of its centriole-associated trichome, which is reminiscent of the translational PCP of monociliated epithelia seen in vertebrates. Ciliary beating in the mouse node and zebrafish floor plate is indicated by red circular arrows.

**Figure 4 cells-13-01403-f004:**
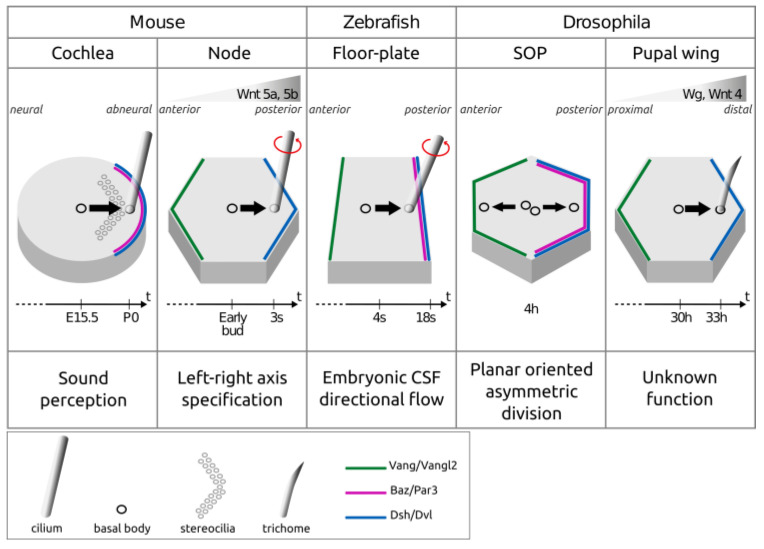
Timing of basal body and centriole planar polarization in different systems. **The planar polarization of basal bodies and centrioles happens within different time scales** ranging from a few hours (*Drosophila* Sensory Organ Precursor (SOP), mouse node and pupal wing) to days (mouse cochlea). This polarization depends on the planar polarization of polarity proteins such as Par3, Vangl2 and Dvl. **Upstream polarizing cues such as Wnt gradients have been described in the mouse node and the *Drosophila* pupal wing**. Centriole position shifts are indicated by black arrows. The bottom line of the table indicates the various functions that depend on the planar polarization of these tissues. E15.5: mouse embryonic day 15.5. P0: mouse post-natal day 0. Early bud: the “bud” stage of a mouse embryo corresponding to E7.5. s: somite; h: hour.

**Figure 5 cells-13-01403-f005:**
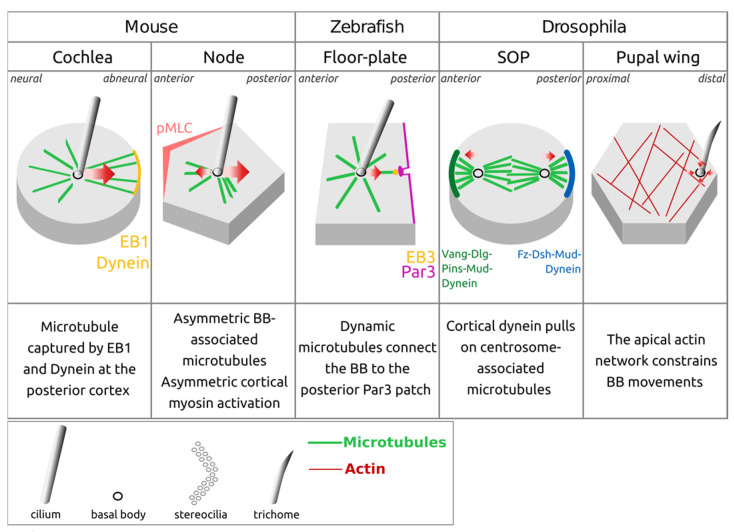
Potential roles of microtubules and actin in basal body and centrosome planar polarization. Microtubules exert a polarized force on basal bodies in several systems, which is likely to be essential for their basal body off-centering. **Microtubule capture coupled with cortical dynein-mediated pulling has been proposed to off-center the basal body** in the mouse cochlea and zebrafish floor plate, a mechanism well described during *Drosophila* SOP polarized division, where anterior Vang-Dlg-Pins-Mud and posterior Fz-Dsh-Mud complexes recruit dynein. In the mouse node, basal body off-centering depends on asymmetry in the basal body associated microtubules and an active non-muscle myosin II pool on the anterior side of the cell apical surface (pMLC: phosphorylated Myosin Light Chain; this phosphorylation indicates active non-muscle myosin II). In *Drosophila* pupal wing cells, the role of microtubules is still unknown, but the apical actin network is required to maintain the centrioles confined to a small region of the apical surface. Red arrows indicate the mechanical forces exerted by the cytoskeleton on centrioles.

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
