# Peer review of "Centriole Translational Planar Polarity in Monociliated Epithelia"

_cells, 2024, doi:10.3390/cells13171403_

Round 1
Reviewer 1 Report
Comments and Suggestions for Authors
I have thoroughly reviewed the manuscript "Centriole translational planar polarity in monociliated epithelia" by Donati et al. I am impressed with the depth of their analysis and the comprehensive citation of previous research on the relationship between primary cilia movement and planar cell polarity in epithelial cells. Their work significantly contributes to our understanding of this complex subject. I am pleased to recommend the acceptance of your manuscript for publication when minor issues are fixed.
Minor points
1)p.7, line181: Comma following "[17]" is highlighted in yellow.
2) p.9 line. 267: Please remove red symbols which are interposing "patches".
Author Response
Minor points:
1) p.7 line 181: Comma following "[17]" is highlighted in yellow.
Done
2) p.9 line 267: Please remove red symbols which are interposing "patches".
Done
Reviewer 2 Report
Comments and Suggestions for Authors
The manuscript entitled Centriole translational planar polarity in monociliated epithelia by Antoine Donati, Sylvie Schneider-Maunoury and Christine Vesque, reviews the molecular mechanisms underlying translational polarity of basal bodies (asymmetric localization) in vertebrate mono-ciliated epithelia, as part of the planar cell polarity (PCP) and their interplay with polarity factors and the cytoskeleton. The review is pertinent, comprehensive and well organized and updates the last developments in the field. I think the manuscript can be published, after revision:
Major suggestions:
· -The description of cilia biogenesis in the beginning of the introduction is not clear and is lacking rigor. For example, “To grow a cilium, the basal body must anchor to a membrane - either the plasma membrane or the ciliary vesicle thanks to a pair of distal appendages (DA).”. Although the subject is not the main target of the review, for those that do not know details about cilia biogenesis this sentence is not informative at all. Moreover, DAs are a complex structure localized in most distal end of mother centriole that can not be referred as a “pair”. If the idea was to say that are two types of appendages at the distal end of mother centriole (a pair) to say a pair of distal appendages is confusing.
Again, the sentence “The basal body is also decorated by sub-distal appendages, among which one basal foot on most motile cilia, and rootlets (Fig 1) [2]” is not very precise. Indeed, basal foot and sub-distal appendages (SDAs) share components, that seem to maintain conserved interactions in both structures, but SDAs composition seems to vary through cell cycle and most probably the architectures of SDAs and basal foot are distinct suggesting that most probably SDAs undergo a remodeling process to originate the basal foot. I suggest that the authors rewrite this part of introduction, even it be summarized.
· -The authors did not mention/discuss nothing about focal adhesion proteins that are found associated with basal bodies and may also play a role in in basal body migration, docking, and spacing. Maybe they can introduce this subject when they discuss the role of actin.
Minor
· -In Figure 1 legend I propose the title: “A motile cilium and its basal body appendages in a monociliated epithelia cell” instead of “A motile cilium and its appendages”.
· -Uniformize through the text drosophila by Drosophila (italic since is the Genus).
· -I suggest that legends of figures must have references or” adapted from” if this is the case.
· A reference, probably [12], should be included in sentence of line 263/264 “The most comprehensive results on Par3 function come from the study of the zebrafish floor-plate (….).
· A Reference is missing in the statement in line 366: “Accordingly, in Madin-Darby canine kidney (MDCK) cells, Rpgrip1l was shown to antagonize Inversin and Nphp4, two transition zone proteins that target Dvl for proteasome-mediated degradation. Thus, Rpgrip1l modulates the PCP pathway by regulating the stability of the core PCP protein Dvl.”
Author Response
Major suggestions:
- The description of cilia biogenesis in the beginning of the introduction is not clear and is lacking rigor. For example, “To grow a cilium, the basal body must anchor to a membrane - either the plasma membrane or the ciliary vesicle thanks to a pair of distal appendages (DA).”. Although the subject is not the main target of the review, for those that do not know details about cilia biogenesis this sentence is not informative at all. Moreover, DAs are a complex structure localized in most distal end of mother centriole that can not be referred as a “pair”. If the idea was to say that are two types of appendages at the distal end of mother centriole (a pair) to say a pair of distal appendages is confusing.
- Again, the sentence “The basal body is also decorated by sub-distal appendages, among which one basal foot on most motile cilia, and rootlets (Fig 1) [2]” is not very precise. Indeed, basal foot and sub-distal appendages (SDAs) share components, that seem to maintain conserved interactions in both structures, but SDAs composition seems to vary through cell cycle and most probably the architectures of SDAs and basal foot are distinct suggesting that most probably SDAs undergo a remodeling process to originate the basal foot. I suggest that the authors rewrite this part of introduction, even it be summarized.
We agree that we oversimplified the description of cilia structure and were not precise enough in our description.
We now specify in the title of Figure 1 that we present the structure of a motile monocilium and explain the differences in the text with motile cilia from multiciliated cells (lines 37-39, page 1). We describe in greater detail the distal and sub-distal appendages of different types of cilia (lines 43-48, pages 1-2) and refer to an appropriate review for the readers (Mahen 2021, ref. 8). We mention that we drew a rootlet that is at the opposite of a basal foot, which is frequent but not mandatory in motile monocilia. We removed the term “appendages pair”, which introduced confusion.
- The authors did not mention/discuss nothing about focal adhesion proteins that are found associated with basal bodies and may also play a role in in basal body migration, docking, and spacing. Maybe they can introduce this subject when they discuss the role of actin.
We have now introduced a short description of focal adhesion proteins that are associated with basal bodies in multiciliated cells and described their functions (revealed in Antoniades 2014, ref. 83) in Paragraph 6.1 (lines 521-525, page 12), opening the possibility that they might also provide a link with the subapical actin cytoskeleton.
Minor points:
- In Figure 1 legend I propose the title: “A motile cilium and its basal body appendages in a monociliated epithelia cell” instead of “A motile cilium and its appendages”.
We changed the title of Figure 1 to « A motile cilium from monociliated epithelia and its appendages »
- Uniformize through the text drosophila by Drosophila (italic since is the Genus).
Done
- I suggest that legends of figures must have references or” adapted from” if this is the case.
The figures are not adapted from previous articles, they were designed by A. Donati with the initial help of S. Gournet (see Acknowledgements).
- A reference, probably [12], should be included in sentence of line 263/264 “The most comprehensive results on Par3 function come from the study of the zebrafish floor-plate (….).
Reference added (now in lines 252-253, ref 18)
- A Reference is missing in the statement in line 366: “Accordingly, in Madin-Darby canine kidney (MDCK) cells, Rpgrip1l was shown to antagonize Inversin and Nphp4, two transition zone proteins that target Dvl for proteasome-mediated degradation. Thus, Rpgrip1l modulates the PCP pathway by regulating the stability of the core PCP protein Dvl.”
Reference added (no in lines 355-358, ref 62)
Reviewer 3 Report
Comments and Suggestions for Authors
I have reviewed the manuscript titled "Centriole Translational Planar Polarity in Monociliated Epithelia" and find the topic highly relevant and timely, offering significant insights into the molecular and cellular mechanisms governing ciliary polarity in vertebrate systems. The review is well written and only requires minor corrections/changes before submission.
General Comments:
- The figures would benefit from more detailed legends that not only describe what is shown but also discuss the relevance of these visualizations to the overall narrative of the manuscript. Additionally, it would be an exciting addition to the manuscript to discuss the potential implications of these insights in biomedical applications, such as in the development of therapeutic strategies for diseases related to ciliary dysfunction in the Conclusion and Perspective section.
- Please clearly separate Figure legends and text
Minor corrections
L 70: the term pericentriolar matrix appears for the first time. Maybe a short description is beneficial for the reader
L 172: I am not familiar with this nomenclature, maybe Vangl1−/−;Vangl2−/− double-mutant is clearer. Same for Dvl mutants in L 175.
L181: remove yellow highlight for comma
L 188: looks like a spacing too much between proteins. The autors.
L206 and 207: remove doted lines from underneath the citations
L268: please remove (“patches”) or the correction in red
L279: I don’t understand this sentence, please rephrase and remove doted lines from citations, the correct term is neighbour or neighbourhood…, it could interact with Fz3a in its anterior neighbor [21,23].
L288: insert “are” between “and” and “important”
L320: “as suggested by two studies which showed that IN Ift88 [50] and Kif3a [51] mouse mutants” please remove “in”
L382: must be coordinated with THE embryonic axis (insert the) and We W must be capital
L383: please add a full stop after “positioning”
L426 and L427: adapt type-face, it appears to be another typeface or size
L630: “…Would the microtubule network be modified? This type of experiments may allow to address translational polarity dependence on polarized basal body…” this is not an experiment but a question. Please rephrase to something like “To experimentally answer these questions may allow to address….”
L684-686: Please remove: “In this section, you can acknowledge any support given which is 684 not covered by the author contribution or funding sections. This may include administrative and 685 technical support, or donations in kind (e.g., materials used for experiments).”
Comments on the Quality of English Language
I have reviewed the manuscript titled "Centriole Translational Planar Polarity in Monociliated Epithelia" and find the topic highly relevant and timely, offering significant insights into the molecular and cellular mechanisms governing ciliary polarity in vertebrate systems. The review is well written and only requires minor corrections/changes before submission.
General Comments:
- The figures would benefit from more detailed legends that not only describe what is shown but also discuss the relevance of these visualizations to the overall narrative of the manuscript. Additionally, it would be an exciting addition to the manuscript to discuss the potential implications of these insights in biomedical applications, such as in the development of therapeutic strategies for diseases related to ciliary dysfunction in the Conclusion and Perspective section.
- Please clearly separate Figure legends and text
Minor corrections
L 70: the term pericentriolar matrix appears for the first time. Maybe a short description is beneficial for the reader
L 172: I am not familiar with this nomenclature, maybe Vangl1−/−;Vangl2−/− double-mutant is clearer. Same for Dvl mutants in L 175.
L181: remove yellow highlight for comma
L 188: looks like a spacing too much between proteins. The autors.
L206 and 207: remove doted lines from underneath the citations
L268: please remove (“patches”) or the correction in red
L279: I don’t understand this sentence, please rephrase and remove doted lines from citations, the correct term is neighbour or neighbourhood…, it could interact with Fz3a in its anterior neighbor [21,23].
L288: insert “are” between “and” and “important”
L320: “as suggested by two studies which showed that IN Ift88 [50] and Kif3a [51] mouse mutants” please remove “in”
L382: must be coordinated with THE embryonic axis (insert the) and We W must be capital
L383: please add a full stop after “positioning”
L426 and L427: adapt type-face, it appears to be another typeface or size
L630: “…Would the microtubule network be modified? This type of experiments may allow to address translational polarity dependence on polarized basal body…” this is not an experiment but a question. Please rephrase to something like “To experimentally answer these questions may allow to address….”
L684-686: Please remove: “In this section, you can acknowledge any support given which is 684 not covered by the author contribution or funding sections. This may include administrative and 685 technical support, or donations in kind (e.g., materials used for experiments).”
Author Response
General Comments:
- The figures would benefit from more detailed legends that not only describe what is shown but also discuss the relevance of these visualizations to the overall narrative of the manuscript. Additionally, it would be an exciting addition to the manuscript to discuss the potential implications of these insights in biomedical applications, such as in the development of therapeutic strategies for diseases related to ciliary dysfunction in the Conclusion and Perspective section.
To better integrate the relevance of the visualization presented in the figures to the overall narrative of the manuscript, we chose to highlight their general message (s) by writing it in bold within the figure legend.
To discuss the potential implications of these insights in biomedical applications, we added in the “conclusion and perspectives” part a paragraph (lines 678-712, pages 15-16) describing genetic evidence for the implication of Wnt-PCP main actors and effectors into various pathologies such as neural tube defects (NTD) and congenital hydrocephaly (CH), that might reflect cilia dysfunctions.
- Please clearly separate Figure legends and text
We have changed the position of the figures within the text, reduced their size, and better separated the figure legends from the text.
Minor corrections:
- L 70: the term pericentriolar matrix appears for the first time. Maybe a short description is beneficial for the reader
Done (lines 84-88)
- L 172: I am not familiar with this nomenclature, maybe Vangl1−/−;Vangl2−/− double-mutant is clearer. Same for Dvl mutants in L 175.
Done
- L181: remove yellow highlight for comma.
Done
- L 188: looks like a spacing too much between The autors.
Done
- L206 and 207: remove doted lines from underneath the citations
Done
- L268: please remove (“patches”) or the correction in red
Done
- L279: I don’t understand this sentence, please rephrase and remove doted lines from citations, the correct term is neighbour or neighbourhood…, it could interact with Fz3a in its anterior neighbor [21,23].
Neighbor is the American spelling for neighbour
- L288: insert “are” between “and” and “important”
We added “which are” to conform with the structure of the sentence.
- L320: “as suggested by two studies which showed that IN Ift88 [50] and Kif3a [51] mouse mutants” please remove “in”
Done
- L382: must be coordinated with THE embryonic axis (insert the) and We W must be capital
Done
- L383: please add a full stop after “positioning”
Done
- L426 and L427: adapt type-face, it appears to be another typeface or size
Typeface adapted
- L630: “…Would the microtubule network be modified? This type of experiments may allow to address translational polarity dependence on polarized basal body…” this is not an experiment but a question Please rephrase to something like “To experimentally answer these questions may allow to address….”
Done
- L684-686: Please remove: “In this section, you can acknowledge any support given which is 684 not covered by the author contribution or funding sections. This may include administrative and 685 technical support, or donations in kind (e.g., materials used for experiments).”
Done
Round 2
Reviewer 2 Report
Comments and Suggestions for Authors
Dear Sirs,
In general, the authors have taken my suggestions into account, therefore I think the manuscript can be accepted for publication in the present form.
Yours sincerely,
Helena Soares